# Serum Myo-Inositol, Dimethyl Sulfone, and Valine in Combination with Creatinine Allow Accurate Assessment of Renal Insufficiency—A Proof of Concept

**DOI:** 10.3390/diagnostics11020234

**Published:** 2021-02-03

**Authors:** Jochen Ehrich, Laurence Dubourg, Sverker Hansson, Lars Pape, Tobias Steinle, Jana Fruth, Sebastian Höckner, Eric Schiffer

**Affiliations:** 1Department of Pediatric Kidney-, Liver- and Metabolic Diseases, Children’s Hospital, Hannover Medical School, 30625 Hannover, Germany; ehrich.jochen@mh-hannover.de; 2Service d’Explorations Fonctionnelles Rénaleset Métaboliques, Hôpital Edouard Herriot, 69437 Lyon, France; laurence.dubourg@chu-lyon.fr; 3Department of Pediatrics, Sahlgrenska University Hospital, 413 45 Gothenburg, Sweden; sverker.hansson@pediat.gu.se; 4Department of Pediatrics II, University Hospital Essen, 45147 Essen, Germany; lars.pape@uk-essen.de; 5Department of Research and Development, numaresAG, 93053 Regensburg, Germany; tobias.steinle@numares.com (T.S.); jana.fruth@numares.com (J.F.); sebastian.hoeckner@numares.com (S.H.)

**Keywords:** kidney function, metabolomics, nuclear magnetic resonance spectroscopy, biomarker

## Abstract

Evaluation of renal dysfunction includes estimation of glomerular filtration rate (eGFR) as the initial step and subsequent laboratory testing. We hypothesized that combined analysis of serum creatinine, myo-inositol, dimethyl sulfone, and valine would allow both assessment of renal dysfunction and precise GFR estimation. Bio-banked sera were analyzed using nuclear magnetic resonance spectroscopy (NMR). The metabolites were combined into a metabolite constellation (GFR_NMR_) using *n* = 95 training samples and tested in *n* = 189 independent samples. Tracer-measured GFR (mGFR) served as a reference. GFR_NMR_ was compared to eGFR based on serum creatinine (eGFR_Crea_ and eGFR_EKFC_), cystatin C (eGFR_Cys-C_), and their combination (eGFR_Crea-Cys-C_) when available. The renal biomarkers provided insights into individual renal and metabolic dysfunction profiles in selected mGFR-matched patients with otherwise homogenous clinical etiology. GFR_NMR_ correlated better with mGFR (Pearson correlation coefficient r = 0.84 vs. 0.79 and 0.80). Overall percentages of eGFR values within 30% of mGFR for GFR_NMR_ matched or exceeded those for eGFR_Crea_ and eGFR_EKFC_ (81% vs. 64% and 74%), eGFR_Cys-C_ (81% vs. 72%), and eGFR_Crea-Cys-C_ (81% vs. 81%). GFR_NMR_ was independent of patients’ age and sex. The metabolite-based NMR approach combined metabolic characterization of renal dysfunction with precise GFR estimation in pediatric and adult patients in a single analytical step.

## 1. Introduction

Although in use for decades, the methods available for estimating glomerular filtration rate (eGFR) with endogenous markers still present important drawbacks [1,2] and thus were described to be a weak link in renal diagnostics [3]. All endogenous filtration markers also have non GFR determinants [1]. Limiting factors include the analytical determination of the substance itself [4,5,6], substances interfering with marker quantification [7], as well as non-glomerular filtration determinants, such as synthesis, tubular reabsorption, secretion, and extra-renal elimination. Numerous equations were developed to compensate for these limiting factors [8] and are now an essential part of routine clinical practice, although still having weaknesses [1]. It was concluded that a single filtration marker is unlikely to successfully overcome the limitations of endogenous metabolites, because of variables affecting the pathophysiology of chronic kidney disease (CKD) other than GFR [1,2,4]. Indeed, combining two markers like creatinine and cystatin C improved the accuracy of GFR estimation [2]. However, the persistence of these limitations prompted Porrini et al. to suggests that the problem perse might be associated with the biochemical natureof creatinine and cystatin C as markers of renal function, rather than with the mathematical methods used for GFR estimation [8]. In addition, Hsu and Bansal suggested that determining actual GFR with utmost accuracy may be a less important goal compared to assessing patients’ complex renal dysfunction and complications according to the stage of CKD [9].

Therefore, we aimed at a more complex approach that interprets multiple biomarkers reflecting both the glomerular filtration rate and CKD-associated renal dysfunction. Such an approach requires the quantification of several renal biomarkers with high precision and accuracy. To avoid increasing analytical costs associated with multiple single biomarker assays, we applied nuclear magnetic resonance spectroscopy (NMR) as a multiplex analyzer capable to precisely quantify multiple unlabeled metabolites in a simultaneous physical measurement step [10].

Recently, numerous metabolomic screens extensively described a multitude of NMR-accessible renal biomarkers [11,12,13,14,15,16,17,18,19]. For a proof of concept, we evaluated the most consistently reported biomarkers with respect to their (patho-)physiological relevance for renal function and/or renal and extra-renal co-morbidities and selected suitable candidates for a targeted analysis. Numerous publications reported an increase of serum myo-inositol levels in CKD with a good inverse correlation with GFR [11,16,17,20,21,22,23,24]. Besides being an essential component of inositol phosphates, which are important second messengers in the cell and are involved in different signaling pathways, myo-inositol is a uremic toxin. We also chose dimethyl sulfone, which is a sulfur-containing substance sensitive to oxidative stress and was found to be elevated in CKD patients [12,14,25,26,27,28,29,30,31]. In addition, we selected valine whose blood levels, unlike most other blood metabolites, correlated positively with eGFR in metabolomics screens and numerous publications reported metabolic acidosis to induce degradation of valine, causing reduced plasma levels of valine in CKD [13,15,17,18,32,33].

We hypothesized that NMR-based analysis of myo-inositol as a marker of uremia, dimethyl sulfone as a marker of oxidative stress, and valine as an indicator of acid-base metabolism in combination with creatinine would provide a ‘metabolite constellation’ that describes the complex renal and metabolic dysfunction in CKD. We tested whether this framework would allow a precise estimation of glomerular filtration.

## 2. Materials and Methods

### 2.1. Cohorts and Samples

For biomarker quantification, bio-banked serum samples from 320 individuals from Lyon (France), Gothenburg (Sweden), and Berlin (Germany) [34] were used. All adult individuals gave informed consent before undergoing GFR measurement. As children were involved in this research activity, their assent and the permission of their parents was obtained. Assent was defined as a child’s affirmative agreement to participate in research. A signed informed consent form from the child as well as from the parents was obtained. The respective institutional review boards covered ethical approval for NMR analysis of the samples in adherence to the Declaration of Helsinki. Descriptive statistics of the study sample are given in Table 1. Samples were stored at −80 °C and underwent not more than one freeze-thaw cycle before central NMR analysis. Fourteen samples had to be excluded due to missing clinical data. In a further 21 samples, the obtained NMR spectra failed quality control criteria even after re-analysis. Outlier analysis showed one sample with extreme discrepancy between the creatinine quantified by NMR and the chemically measured value and was excluded. The remaining sample set of 284 samples was split in a training set of *n* = 95 samples for bio-statistical modeling and an independent test set of *n* = 189 samples for performance evaluation (Appendix A).

### 2.2. Benchmarking

Serum creatinine-based and cystatin C-based eGFR served as benchmarks. eGFR equations were selected according to KDIGO recommendations [35]. Serum creatinine-based eGFR (eGFR_Crea_) was calculated using the CKD-EPI 2009 creatinine equation [36], while for pediatric patients, the updated “Bedside” Schwartz formula [37] was used. For an age-independent serum creatinine-based eGFR, the European Kidney Function Consortium equation [38] was applied (eGFR_EKFC_). The CKD-EPI 2012 cystatin C equation [2] and the cystatin C-based equation derived from the CKiD cohort [39] were used for calculating eGFR from cystatin C (eGFR_Cys-C_) in adults and children, respectively. For the adult subset, the 2012 CKD-EPI creatinine-cystatin C equation [2] was applied to calculate eGFR from both creatinine and cystatin C (eGFR_Crea-Cys-C_). Patients aged 18 years or older were considered adults.

### 2.3. mGFR, Serum Creatinine, and Cystatin C Measurements

The study samples had a mixture of inulin [40], iohexol [41], or ^51^Cr-EDTA [42] GFR measurements as the reference standard. The results were expressed per 1.73 m^2^-body surface according to the Dubois equation: body surface area = height^0.725^ × weight^0.425^ × 0.007184. Applied mGFR methods were reported to have sufficient accuracy compared with the inulin method [43]. All creatinine measurements were performed with methods traceable to the National Institute of Standards and Technology and were isotope-dilution mass spectrometry calibrated [44]. Serum cystatin C measurements of the Berlin cohort were measured at Labor Limbach Heidelberg (Heidelberg, Germany) using the PENIA N Latex^®^ assay on the BN™ II System (Siemens Health Care Diagnostics, ex-Dade-Behring, Marburg, Germany). For samples from Lyon and Gothenborg with sufficient leftover volume, cystatin C was measured with the Human Cystatin C ELISA from Biovendor (BioVendor—Laboratornimedicinaa.s., Brno, Czech Republic) calibrated to standard reference material ERM-DA471/IFCC at Laborarztpraxis van de Loo, Schwäbisch Gmünd, Germany.

### 2.4. NMR Analysis

Serum was thawed at room temperature and 630 µL were mixed with 70 µL of Axinon^®^ serum additive solution (numares AG, Regensburg, Germany). A total of 600 µL were transferred to 5-mm NMR tubes. Runs were carried out in batches of up to 93 samples, including a calibration sample and two process controls. Samples were pre-heated at 37 °C for 7.5 min before NMR measurement in a Bruker Avance III 600 MHz and a 5-mm PATXI probe equipped with automatic Z gradients shimming (Bruker Corporation, Billerica, MA, USA). A modified version of the CPMG pulse sequence was used as described [45]. The sequence accounts for a rapid and periodic refocusing of the J evolution by coherence transfer. A 90-degree pulse was inserted at the midpoint of a double spin echo, leading to refocusing and consequent quenching of homonuclear J modulation. -NMR spectra were recorded using a spectral width of 20 ppm, with a recycling delay of 1.5 s, 16 scans, and a fixed receiver gain of 50.4. A cycling time d2 of 8 ms was used together with a corresponding T2 filter of 112 ms. The mixing time τ between two consecutive spin echoes was 400 µs. NMR data were phase and baseline corrected by algorithms developed in-house and using the lactate doublet at 1.32 ppm as the reference. Spectra underwent automatic processing and quality control as part of the magnetic group signaling^®^ technology based on the offset and slope of the baseline in selected spectral regions and selected signals, e.g., position, shape, and width. The system allows detection and flagging of spectra of insufficient quality and includes a calibration for scaling to make spectra comparable across runs and devices.

### 2.5. Biomarker Quantification

For robust quantification, we curve-fitted pseudo-Voigt profiles to creatinine, myo-inositol, dimethyl sulfone, and valine NMR signals. This method allows determination of the goodness of fit by assessing differences between the spectrum and fitted profile, and thus indicates when the quantification is unreliable due to interference by other metabolites. For analytical validation of biomarker quantification, precision, linearity, and bias were analyzed.

#### 2.5.1. Precision

For analytical validation of biomarker quantification, a 50-mL serum pool of 25 pediatric outpatients between 8 and 17 years of age with eGFR_Crea_ < 90 mL/min/1.73 m^2^ at the day of serum collection was collected between October 2016 and January 2017 at Hannover Medical School (INPREM cohort). Sampling was ethically approved by Hannover Medical School’s institutional review board (No. 3396-2016, dated 15 September 2016). Individual serum samples were stored no longer than 2 h at room temperature before interim storage at −20 °C. Long-term storage was at −80 °C. Samples were pooled just before preparation for NMR analysis. Additional human serum pools were purchased by Bavarian Red Cross (Regensburg, Germany) and stored at −20 °C until utilization. Analytical precision was assessed using three distinct serum pools: the first two pools consisted of commercially available Bavarian Red Cross (adult normal GFR, adult low adult GFR) serum, while the third pool contained pooled sera collected from 25 pediatric patients enrolled in the INPREM study (pediatric low GFR, see above for details). Serum pools were measured in five runs with three replicates per measurement on a single NMR device, resulting in 15 NMR measurements for each pool. Within-run, between-run, and total variation were analyzed using a fully nested model II ANOVA, and analytical coefficients of variation (CV) were computed.

#### 2.5.2. Linearity

In the linearity study, a working range for each biomarker was determined within which the relationship between the observed values and the true concentrations of the metabolite of interest is linear. Therefore, a serum pool with high metabolite serum levels (i.e., high pool) was prepared by spike-in before preparing linearity samples. A linear dilution in 13 concentration steps down to 0% of the high pool was prepared. For each concentration step, three replicates were analyzed.

#### 2.5.3. Bias

The bias analysis measured the closeness between NMR-measured metabolite concentrations and spike recovery or whenever available clinical chemistry reference values. For dimethyl sulfone, myo-inositol, and valine, bias was determined by spike recovery experiments. To generate the samples, non-modified and dialyzed sera were pooled to generate ‘mini-pools’. Additives spiked with the respective metabolites were generated for 12 different concentration levels. The respective controls consisted of the same mini-pool prepared with additives, but without the respective metabolite. In total, 138 mini-pools were used in the spike recovery experiment. For creatinine, standard clinical chemistry reference methods were used (Creatinine reagent OSR6678 on a Beckman Coulter AU640 analyzer, Beckman Coulter Inc., Brea, CA, USA). In total, 120 human serum mini-pools, partly spiked or dialyzed to cover a broader concentration range, were measured in duplicates by NMR and by the reference method. In total, 115 mini-pools were used in the study.

### 2.6. GFR Modeling 

All statistical analyses were carried out with R software v3.5.1 [46]. In order to find a quantitative relation between the biomarkers and mGFR, a linear regression model was set up using second-order interactions of the form:(1)$$y=\beta_{0}+\underset{i}{\sum }\beta_{i}x_{i}+\underset{j=1}{\sum }\underset{k=j+1}{\sum }\beta_{jk}x_{j}x_{k}+\varepsilon$$
where *y* represents the target value, $x_{i}$ is the input variables, $\beta_{i}$ is the corresponding model coefficients, *β*_0_ is the intercept, and *ε* denotes the error term. The quantified renal biomarker levels were transformed to their natural logarithmic value. The cost function within the coefficient estimation determines the way in which values of different ranges are weighted. As all metabolites showed range-dependent association with mGFR, fitting of separate local submodels for different mGFR ranges was applied. Therefore, the mGFR range was divided into two sections, i.e., <90 (‘low GFR’ and >60 mL/min/1.73 m^2^ ‘high GFR’) with an overlapping transition area between 60 and 90 mL/min/1.73 m^2^. To obtain a result for the NMR-based GFR, an interpolation function was applied depending on the distance to the mGFR limits 0 and 150:(2)$${\hat{y}}_{combined}=\left\{ \begin{array}{ll} {\hat{y}}_{low} & {\hat{y}}_{low}<{0,\hat{y}}_{high}\leq 150,\\ {\hat{y}}_{high} & {\hat{y}}_{high}>150,\\ \frac{{\left(150-{\hat{y}}_{high}\right)}^{p}}{{{\hat{y}}_{high}}^{p}+{\left(150-{\hat{y}}_{high}\right)}^{p}}{\hat{y}}_{low}+\frac{{\hat{y}}_{low}}{{{\hat{y}}_{low}}^{p}+{\left(150-{\hat{y}}_{high}\right)}^{p}}{\hat{y}}_{high} & else. \end{array}\right.$$
where ${\hat{y}}_{high}$ and ${\hat{y}}_{low}$ are the prediction values of the lower and the upper submodel, and *p* is a power parameter. A value of *p* = 4 was used. For modelling, a maximal model size of five features (four substances plus one interaction for each local submodel) was allowed.

### 2.7. Model Selection

To find the most suitable regression model, we performed a hundred times repeated five-fold cross-validation for each substance combination. Model stability was maximized by removing all substance combinations with a coefficient of variation of the cross-validated substance coefficients above 15% to 20% depending on the applied model approach. The substance combinations were then selected by their performance on root mean square error (RMSE), mean absolute error (MAE), and logarithmic RMSE (RMSLE) in the different mGFR ranges. The lower part of the GFR range was best estimated by using log-log-regression, whereas the upper part was predicted best with linear regression.

## 3. Results

### 3.1. Biomarker Quantification

The development of quantification algorithms together with quality control strategies enabled effective quantification of creatinine, myo-inositol, dimethyl sulfone, and valine by NMR with limits of quantification of approximately 10–20 µmol/L. Table 2 depicts an overview of the analytical validation results obtained for the respective quantifiers. Total analytical precision (within and between run) for three different serum pools (normal GFR, low adult GFR, and low pediatric GFR), linearity, and trueness are covered. Total analytical imprecision for all markers was below 15%. Imprecision increased for serum levels at limits of quantification of approximately 10 µmol/L. For all metabolites, Pearson correlation was r > 0.99 except for dimethyl sulfone with r > 0.98. The analytical performance of the NMR platform allowed a sensitive, specific, precise, and accurate measurement of the serum biomarker levels over a linear range covering both physiological and pathophysiological concentration ranges.

### 3.2. GFR Estimation

We tested the hypothesis of whether adding myo-inositol, valine, and dimethyl sulfone to serum creatinine would allow accurate estimation of mGFR. Serum concentrations of creatinine, myo-inositol, and dimethyl sulfone were negatively correlated, with mGFR with log-log Pearson correlation coefficients of −0.783, −0.612, and −0.520, respectively. In contrast, serum valine was positively correlated with mGFR. Interestingly, this weak positive correlation with mGFR (r = 0.206, 97.5% CI−0.04 to 0.43) increased to r = 0.534 (97.5% CI 0.33 to 0.68), when valine was correlated with the residual variance of mGFR that contributions of creatinine and myo-inositol alone are unable to cover (Figure 1).

Exhaustive searches of all possible combinations of the four metabolites were carried out in the training set comprising *n* = 95 samples using mGFR as the reference. The most suitable model took the form:*ŷ_low_* = e^−1.34^ × creatinine^−0.83^ × myo-inositol^2.22^ × valine^0.54^ × creatinine^−0.57 log(myo-inositol)^,(3)
*ŷ_high_* = 285.00 − 38.35 × log(creatinine) − 12.47 × log(dimethyl sulfone),(4)
with metabolite concentrations in µmol/L.

Total analytical precision (within and between run) for GFR_NMR_ (normal GFR, low adult GFR, and low pediatric GFR) was between 4.2% and 9.5% (Table 2). The performance of GFR_NMR_ in estimating mGFR was assessed in the independent test set (*n* = 189) and the percentage of estimated GFR values within 30%, 15%, and 10% of mGFR values (P_30_, P_15_, and P_10_) were calculated (Table 3).

Compared to the creatinine-based equations, i.e., CKD-EPI for adults and Schwartz Bedside for children, GFR_NMR_ showed a Pearson correlation coefficient of r = 0.84 compared to r = 0.79, and a 35% reduction in the overall root mean square error (RMSE 16.5 vs. 25.3, Figure 2a,c and Table 3). GFR_NMR_ showed a P_30_ of 81% (CKD stages 5–3: 72%, CKD 2: 86%, CKD 1: 88%) compared to 64% (CKD stages 5–3: 46%, CKD 2: 68%, CKD 1: 86%) observed for eGFR_Crea_. For P_15_ as well as for P_10_, consistent improvements independent of the CKD stages were observed (Table 3). For the creatinine-based European Kidney Function Consortium (eGFR_EKFC_) equation, a Pearson correlation coefficient of r = 0.80, an RMSE of 19.6, and a P_30_ value of 74% were observed (Figure 2b and Table 3).

In the subset of samples for which cystatin C values were available, the biomarker constellation outperformed cystatin C-based equations (CKD-EPI for adults and the CKiD-derived equation for children) with a better correlation with mGFR (r = 0.86 vs. 0.66), and a P30 value of 81% compared to 72% for eGFR_Cys-C_ (Figure 3a,b and Table 4). Moreover, GFR_NMR_ showed a similar P_30_ value in adults compared to the combined CKD-EPI equation, which uses both creatinine and cystatin C as variables (81% vs. 81%, Figure 3c,d and Table 4). Consequently, GFR_NMR_ matched or even exceeded the performance of the eGFR equations currently recommended by KDIGO.

### 3.3. Subgroup Analysis of GFR_NMR_ According to Sex and Age

In the test set of *n* = 189 serum samples, we tested in multivariate variance analysis whether the accurate estimation of mGFR using the NMR approach is independent of the age and sex of patients. The individual serum levels of creatinine, myo-inositol, valine, and dimethyl sulfone were agedependent with *p*-values < 0.0001, but independent of patients’ sex (*p* = 0.685, 0.548, 0.270, and 0.243, respectively). However, when the individual biomarkers were considered as a metabolite constellation, an agedependency was no longer observed. GFR_NMR_ was a highly significant predictor of mGFR and neither age nor sex significantly improved the mGFR estimation (Table 5).

### 3.4. Molecular Phenotyping by Matched Sample Sets

Biomarker profiling was applied to two sets of three age- and mGFR-matched male patients from the test set with CKD stage 2 (mGFR of 62 mL/min/1.73 m^2^ (set 1) or 78 mL/min/1.73 m^2^ (set 2)) during end-stage liver disease (Figure 4). This procedure aimed at testing the hypothesis of whether the chosen four biomarkers may provide further insight into CKD pathophysiology in these patients by characterizing renal and metabolic dysfunction. In order to compare the obtained metabolite profiles, measured biomarker concentrations were transformed into z-scores to enable a direct comparison of the observed fold-changes from one marker to the other. The obtained z-scores were plotted in a chart with one axis for creatinine (as a primary marker of filtration), a second for dimethyl sulfone (as a marker of oxidative stress), a third axis for myo-inositol (as a uremic toxin), and a fourth axis for valine (as a marker of acid-base metabolism, Figure 4).

The obtained biomarker constellations for the three patients with measured GFR of 62 mL/min/1.73 m^2^ (set 1, Figure 4a) differed significantly, although patients were matched for etiology, age, and measured GFR. Patients depicted in red and blue showed above average z-scores for creatinine and valine, while the patient depicted in green had below average z-scores for creatinine, dimethyl sulfone, myo-inositol, and valine. This observation would be in line with the conclusion that the patient depicted in green had only minimal levels of oxidative stress, whereas patients in red and blue showed average or increased levels. 

The three matched patients with measured GFR of 78 mL/min/1.73 m^2^ (set 2) had very similar and average levels of oxidative stress (Figure 4b). The patient depicted in blue showed a higher level of valine when compared to the other two matched patients depicted in green and red, suggesting an absence of metabolic acidosis. In addition, his lower level of myo-inositol argued against the presence of uremia. None of the three patients showed increased levels of oxidative stress indicated by dimethyl sulfone. 

These observations suggest that the set of renal biomarkers bears the potential for molecular phenotyping, providing further insights into individual renal and metabolic dysfunction profiles.

## 4. Discussion

The presented proof of concept shows that the uremic toxin myo-inositol, valine as an indicator of acid-base metabolism, and dimethyl sulfone as a marker of oxidative stress in combination with serum creatinine reflect the glomerular filtration rate as well as CKD-associated renal dysfunction. The novel approach matched or even exceeded the accuracy of eGFR equations currently recommended by KDIGO. In addition, the framework of the four metabolites bears the potential for individualized metabolic phenotyping of CKD patients, providing additional insights into the underlying renal disease and (extra-) renal co-morbidities.

A crucial question with respect to the presented approach of four biomarkers refers to the explanation of why myo-inositol, valine, and dimethyl sulfone effectively complemented serum creatinine in such a way that a sex-independent GFR estimation over a large range of age becomes possible. As tubular excretion of serum creatinine is counterbalanced by tubular re-absorption, estimation of glomerular renal function should also consider tubular dysfunction, which may lead to interstitial fibrosis and induces tubulo-glomerular cross talk [47]. In renal dysfunction, the tubular cells develop several disturbances of metabolism and molecular transport as well as of inflammatory reactions detectable in blood circulation that are reflected only partially by serum creatinine [28,48]. Accordingly, tubular clearances of secretory solutes were suggested to provide complementary information about kidney health beyond measurements of glomerular function alone [49]. Thus, the pathogenesis of renal dysfunction can be considered a complex multi-factorial series of molecular events associated with alterations of various disease pathways. 

Keeping creatinine as a marker may be criticized because eGFR_Crea_ is influenced by age, sex, muscle mass, and other patient factors [1]. For example, current eGFR equations often correct for differences in serum creatinine generation among males and females by ‘dividing’ serum creatinine by the mean or median serum creatinine for males and females [36]. Our concept does not ‘correct’ for sexdifferences. However, the three markers as a whole turned out to be a sufficient set of markers to complement serum creatinine for accurate GFR estimation and outbalanced its deficiencies for admittedly hitherto unknown reasons. The observed values for P_30_ and P_15_ of 0.81 and 0.52, respectively, even fulfill the criteria for sufficient precision as proposed by Soveri et al. [43], i.e., P_30_ ≥ 0.80 and P_15_ ≥ 0.50. Therefore, the complex interplay of the four metabolites, complementing each other in way of mitigating individual weaknesses and potentiating their contribution to overall clinical value, defines a kind of ‘metabolite constellation’. Like individual stars in a star constellation contribute to this constellation’s overall appearance, the concept of the metabolite constellation expands the previous approach suggested by Levey and co-workers to combine metabolites into a panel to more closely correlate with mGFR [19,50], by additionally reflecting CKD-associated renal dysfunction and co-morbidities.

If using GFR alone, different CKD stages are diagnosed in isolation from associated extra renal and metabolic comorbidities (KDIGO guidelines, https://kdigo.org/guidelines, and [51]). Our proof of concept expands this mono-causal approach by interpreting GFR in the context of CKD-associated metabolic co-morbidities. In future, this might offer several advantages compared to standard GFR methods. Firstly, such a test would provide further insights into underlying renal co-morbidities in individual patients even in homogenous clinical etiologies. To fully exploit such a kind of phenotyping approach, the understanding of the effects of biological variation and extra-renal comorbidity on the biomarkers may benefit from hemodynamic or metabolic stimulation tests. These might reveal secondary renal and extra-renal functional responses to a decrease of GFR under variable life conditions. In addition, the current literature-based assignment of the biomarkers to renal and extra-renal pathophysiology needs experimental proof.

The new concept might push the current limits of accuracy for GFR estimation largely by adding further biomarkers to the metabolite constellation. However, significantly enlarged samples sets are needed to implement and to test such expansions due to an increasing risk of overfitting associated with every new term added to the regression equation. Finally, the novel concept bears the potential of accurate estimation of GFR in pediatric, adolescent, adult, and geriatric patients irrespective of the patient’s sex, which would enable GFR monitoring from the age of three years into late adulthood.

Our concept is associated with several strengths and weaknesses. The total number of patient samples of both the training and test cohort would certainly benefit from additional samples. Besides increasing statistical power, validation of the concept in further cohorts, including African-American and Asian ethnic groups, as well as patients with, e.g., type 2 diabetes mellitus under metformin treatment, nephrotic syndrome, or various tubulopathies, would allow a comprehensive evaluation of the potential clinical utility of the method. In addition, our training cohort consisted of a sample set with a heterogeneous reference standard with a mixture of inulin, ^51^Cr-EDTA, or iohexol renal clearances. As even inulin clearance is associated with a coefficient variation of 7% for repeated measurements [52], imprecision might increase even more when renal clearances of ^51^Cr-EDTA or iothalamate and plasma clearances of ^51^Cr-EDTA or iohexolare applied for measuring GFR [43]. Hence, the errors of inulin and other exogenous clearance markers are often underestimated when they are used as referenced standards for establishing new eGFR equations [9]. Although we could not determine any dependency of the GFR_NMR_ results from the applied reference method in post-hoc analysis, we cannot exclude the possibility of a reference or selection bias.

Our results obtained for eGFR equations considering cystatin C might have been influenced by both the prolonged storage times of our bio-banked samples and the use of different ELISA assays for cystatin C quantification. Although sample storage was at −80 °C and the applied assays were calibrated to standard reference material, future work should consider an optimized design. Finally, we established the method on serum samples of at least a 630-µL volume, and its transferability to lower volumes or blood plasma cannot be considered as simply given. However, this may be less a limitation on its ability to perform in clinical routine than its application in clinical research with bio-banked serum samples.

Concerning the strengths of our concept, we employed a technically advanced and standardized analytical platform with proven compatibility for daily diagnostic routine application and worldwide availability [53]. Moreover, we demonstrated its sufficient analytical reproducibility for simultaneous quantification of multiple metabolites with known and unknown correlations with renal dysfunctions. This finding effectively translates the requirements for the application of NMR in clinical routine settings proposed by Markley et al. [53]. Hence, in public health systems under a constantly increasing financial burden, our concept might help to control the incremental costs associated with the step-by-step quantification of single biomarkers as constituents of multi-biomarker panels [54] that have been increasingly proposed in recent years [1]. Finally, we demonstrated the accurate estimation of GFR in a multi-center cohort with the gold standard of renal clearance methods as a reference. The observed accuracy matched or even exceeded the one of serum creatinine, serum cystatin C, or their combination in patients with various nephrological conditions.

In conclusion, we developed and tested a metabolite-based serum test for accurate estimation of GFR in pediatric, adult, and geriatric patients, obviating the need for invasive tracer application and bearing the potential of metabolic phenotyping of CKD patients.

## 5. Patents

Tobias Steinle, Jana Fruth, and Eric Schiffer have a patent application DE-216820.2 pending.

## Figures and Tables

**Figure 1 diagnostics-11-00234-f001:**
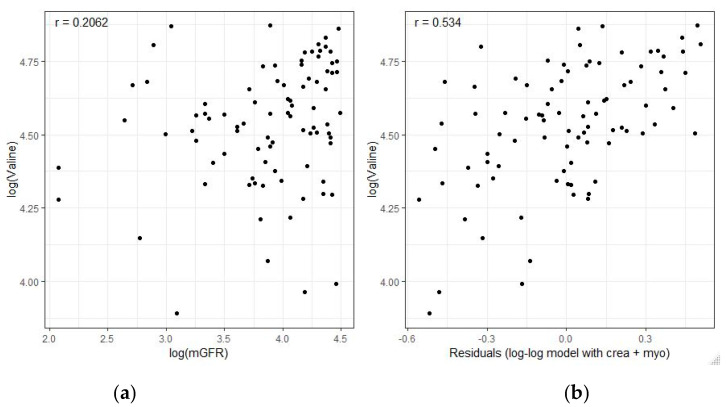
Correlation of valine with measured GFR (mGFR). (**a**) Univariate scatterplot and Pearson correlation coefficient r between the logarithmic serum level of valine and logarithmic mGFR. (**b**) The differences between mGFR and the predicted GFR values by the lower submodely_low_ without valine (consisting only of creatinine in combination with myo-inositol) were interpreted as residuals. Each data point is one residual. When serum valine concentrations were plotted against theses residuals, the correlation coefficient significantly increased to r = 0.534 (97.5% CI 0.33 to 0.68). Hence, serum valine correlates with the residual variance of mGFR that contributions of creatinine and myo-inositol alone are unable to cover.

**Figure 2 diagnostics-11-00234-f002:**
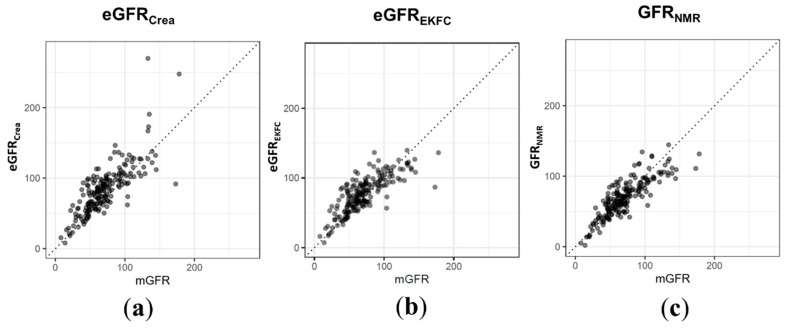
Performance of standard eGFR equations based on serum creatinine in the independent test set compared to the performance of the NMR biomarker constellation. For *n* = 189 patient samples, serum creatinine-based eGFR was calculated (**a**) using the CKD-EPI 2009 creatinine equation for adults and the updated Schwartz Bedside formula for pediatric patients or (**b**) using the EKFC equation. (**c**) GFR_NMR_ correlated better with mGFR (r = 0.84 vs. 0.79 vs. 0.80) and showed a 35% and 16% reduction in RMSE (16.5 vs. 25.3 vs. 19.6) compared to eGFR_Crea_ and eGFR_EKFC_. Overall P_30_ values were more accurate in GFR_NMR_ than in eGFR_Crea_ and eGFR_EKFC_ (81% vs. 64% vs. 74%).

**Figure 3 diagnostics-11-00234-f003:**
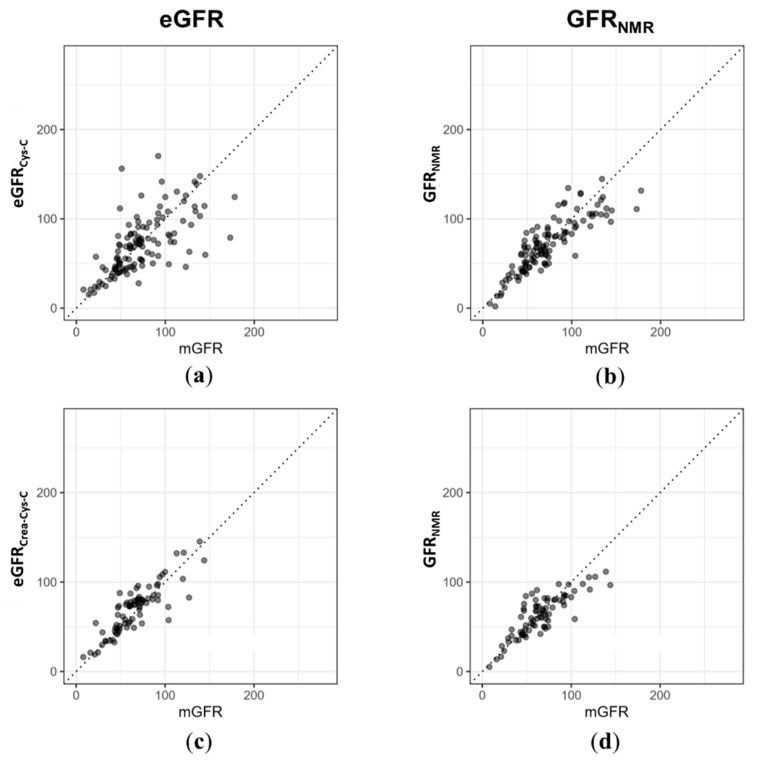
(**a**) For *n* = 118 out of the *n*= 189 test set samples, serum cystatin C data was available. For those, the CKD-EPI 2012 cystatin C equation and the cystatin C-based equation derived from the CKiD cohort were used for calculating cystatin C-based eGFR (eGFR_Cys-C_) in adults and children, respectively. (**b**) GFR_NMR_ correlated better with mGFR (Pearson correlation coefficient r = 0.86 vs. 0.66) and P_30_ values tended to be more accurate in GFR_NMR_ than in eGFR_Cys-C_ (81% vs. 72%). (**c**) For the adult subset of the test cohort with both serum creatinine and cystatin C values available (*n* = 79 of *n* = 189), the 2012 CKD-EPI creatinine–cystatin C equation was applied to calculate eGFR from both creatinine and cystatin C (eGFR_Crea-Cys-C_). (**d**) The Pearson correlation coefficient for GFR_NMR_ was r = 0.83 vs. 0.86 eGFR_Crea-Cys-C_. P_30_ values were 81% vs. 81% for GFR_NMR_ and in eGFR_CreaCys C_, respectively.

**Figure 4 diagnostics-11-00234-f004:**
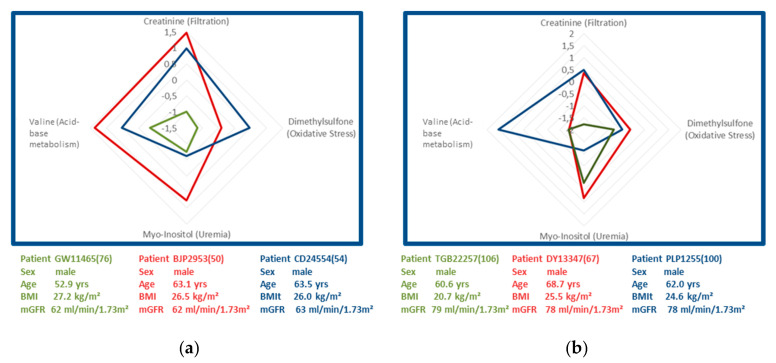
Charts of two sets of three age-, sex-, and measured GFR-matched male patients with end-stage liver disease with CKD stage 2. (**a**) Mean mGFR were 62 mL/min/1.73 m^2^ (set 1) and (**b**) 78 mL/min/1.73 m^2^ (set 2), respectively. Measured biomarker concentrations were transformed into z-scores indicating the plus and minus standard deviation of substance concentrations from the mean of the total cohort. The obtained z-scores were plotted in a chart with one axis for creatinine (as a primary marker of filtration), a second for dimethyl sulfone (as a marker of oxidative stress), a third axis for myo-inositol (as a marker of uremia), and a fourth axis for valine (as a marker of acid-base metabolism).

**Table 1 diagnostics-11-00234-t001:** Descriptive statistics of the study sample.

	Training Set	Test Set
N	95	189
Age (years, range)	4–76	3–88
Age (years, mean ± SD)	34 ± 23 ^a^	51 ± 24 ^a^
Sex (% male)	53	60
mGFR		
range	5–147	8–178
mean ± SD	75 ± 35	71 ± 31
iohexol	54	101
inulin	22	60
^51^Cr-EDTA	19	28
CKD stage		
1	34	49
2	27	70
3	22	57
4	9	11
5	3	2
Storage ^#^ time (years, range)	0.4–13.4	0.4–13.3
Storage ^#^ time (mean ± SD)	2.5 ± 3.6	4.2 ± 3.9

^#^ Storage at −80 °C. ^a^: *p*-value U test < 0.001.

**Table 2 diagnostics-11-00234-t002:** Analytical performance of biomarker quantification.

	Precision						Linearity		Trueness
	Adult Low		Pediatric Low		Adult Normal				
	Mean (µmol/L)	CV (%)	Mean (µmol/L)	CV (%)	Mean (µmol/L)	CV (%)	Low (µmol/L)	High (µmol/L)	Pearson Correlation
creatinine	189.1	6.4	108.3	6.2	107.9	7.0	21	928	0.993
dimethyl sulfone	12.6	13.4	12.2	19.8	8.5	20.7	4	90	0.983
myo-inositol	110.5	11.1	78.5	11.2	68.5	14.2	39	441	0.991
valine	418.0	1.5	310.6	1.9	437.8	3.8	27	1250	0.998
GFR_NMR_	43.5	9.5	68.3	6.0	82.4	4.2	n.a.	n.a.	0.84

Precision is determined in three different pooled sera (adult low GFR, pediatric low GFR, and adult normal GFR). For linearity, upper and lower limits of the linear range are given. For dimethyl sulfone and myo-inositol, trueness was determined by spike recovery. For creatinine and valine, clinical chemistry methods were used.

**Table 3 diagnostics-11-00234-t003:** Diagnostic accuracy. GFR_NMR_ compared to eGFR_Crea_ and eGFR_EKFC_ against mGFR reference in the independent test set of *n* = 189 patient samples.

	eGFR_Crea_	eGFR_EKFC_	GFR_NMR_
RMSE	25.30	19.61	16.51
Pearson correlation	0.79 (0.73–0.84)	0.80 (0.74–0.84)	0.84 (0.80–0.88)
P_10_	0.27 (0.20–0.35)	0.37 (0.29–0.45)	0.35 (0.27–0.43)
P_15_	0.40 (0.32–0.49) ^a^	0.50 (0.41–0.58)	0.52 (0.44–0.60) ^a^
P_30_	0.64 (0.56–0.72) ^b^	0.74 (0.66–0.81)^.^	0.81 (0.74–0.87) ^b^

Note: Serum creatinine-based eGFR_Crea_ was calculated using the CKD-EPI 2009 creatinine equation, while for pediatric patients, the updated Schwartz Bedside formula was used. Numbers in parentheses denote 97.5% confidence intervals, ^a^: *p*-value McNemar test < 0.05, ^b^: *p*-value McNemar test < 0.001.

**Table 4 diagnostics-11-00234-t004:** (**a**) Cystatin-C and (**b**) combined creatinine and Cystatin-C based GFR estimation.

(a)	eGFR_Cys-C_ *	GFR_NMR_	(b)	eGFR_Crea-Cys-C_ **	GFR_NMR_
*n*	118	118	*n*	79	79
RMSE	27.59	17.60	RMSE	14.87	15.32
Pearson correlation	0.66 (0.54–0.75)	0.86 (0.80–0.90)	Pearson correlation	0.86 (0.79–0.91)	0.83 (0.75–0.89)
P_10_	0.28 (0.19–0.38)	0.31 (0.22–0.42)	P_10_	0.42 (0.29–0.55)	0.37 (0.25–0.50)
P_15_	0.40 (0.30–0.51)	0.52 (0.41–0.62)	P_15_	0.57 (0.44–0.69)	0.58 (0.45–0.71)
P_30_	0.72 (0.62–0.81)	0.81 (0.72–0.89)	P_30_	0.81 (0.69–0.90)	0.81 (0.69–0.90)

Note: * CKD-EPI 2012 cystatin C equation and the cystatin C-based equation derived from the CKiD cohort were used for calculating cystatin C-based eGFR (eGFRCys-C) in adults and children. ** The 2012 CKD-EPI creatinine–cystatin C equation was applied. Numbers in parentheses denote 97.5% confidence intervals.

**Table 5 diagnostics-11-00234-t005:** ANOVA results for the model mGFR = GFR_NMR_ + Age + Sex for the test set.

	Df	Sum Sq	Mean Sq	f Value	*p* Value
GFR_NMR_	1	126,206	126,206	464	<0.0001
Age	1	518	518	1.90	0.17
Sex	1	400	400	1.47	0.23
Residuals	185	50,298	272		

Note: Df indicates degrees of freedom, Sum Sq indicates sum of squares, Mean Sq indicates Mean Squares.

## Data Availability

Additional data are provided in the Appendix A.

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
