# Peer review of "Serum Myo-Inositol, Dimethyl Sulfone, and Valine in Combination with Creatinine Allow Accurate Assessment of Renal Insufficiency—A Proof of Concept"

_diagnostics, 2021, doi:10.3390/diagnostics11020234_

Round 1
Reviewer 1 Report
The manuscript by Ehrich et al. want to provide a new approach to evaluate the GFR using new biomarker measured with NMR. The article is well written with some typos and the methods are in the standard of this kind of work, but I have two major concerns.
First, what is the interest of the approach, it is very unlikely that NMR measure could be use in clinical practice compared to the measure of creatinine. So I really don't understand the utility of the approach. In addition the GFR using NMR do not improve the accuracy of evaluation of GFR compared to cys-creat GFR so I really see no interest to this work.
Second, the authors use CKD-EPI as reference equation, but a new equation was recently described, European Kidney Function Consortium (EKFC) equation, and look similar to their results in term of p30. In my opinion the authors must compared their results to this new equations, to really confirm the interest of their approach. the reference of the article is https://doi.org/10.7326/M20-4366
Author Response
The manuscript by Ehrich et al. want to provide a new approach to evaluate the GFR using new biomarker measured with NMR. The article is well written with some typos and the methods are in the standard of this kind of work, but I have two major concerns.
First, what is the interest of the approach, it is very unlikely that NMR measure could be use in clinical practice compared to the measure of creatinine. So I really don't understand the utility of the approach. In addition the GFR using NMR do not improve the accuracy of evaluation of GFR compared to cys-creat GFR so I really see no interest to this work.
The reviewer is pointing out a central challenge that represents a major hurdle for all technical inventions in the medical and clinical field: What is the clinical utility of it and how can accessibility to its routine application be achieved? The answers to these questions make an invention an innovation. As the title of our manuscript indicates, the article does not report a full-fletched product development of a new biomarker test ready for routine application. We rather report results of a ‘proof of concept’ project laying the base for the initiation of such developmental work. This should be kept in mind when reading our point-to-point reply to the appreciated comments.
The AXINON® NMR platform that we applied in the study is already in clinical routine application for the quantification of lipoprotein particles, such as LDL-p, for cardiometabolic risk assessment in the US. Major laboratories operating at the national level, such ARUP (www.aruplab.com), Boston Heart Diagnostics (www.bostonheartdiagnostics.com) or Mayo Clinic Laboratories (www.mayocliniclabs.com), demonstrate with more than two million successful analyses that the technology can be applied with economically attractive reimbursement models in clinical routine settings. The application of LDL-p as a biomarker is to a certain degree comparable to the GFR metabolite constellation: LDL-p is applied as an additional parameter in use-cases where simple cholesterol measurements are of limited value. As we already point out in our manuscript there exist multiple clinical uses-cases were simple creatinine measurement cannot provide the clinician with sufficient information. A future GFRNMR test might be a valuable adjunct to simple creatinine measurement in such use-cases. GFRNMR could (1) provide more accurate GFR estimation in special use-cases independent of patients’ age, (2) offer more detailed pathophysiological insights by the described molecular phenotyping approach, and (3) might even supplement creatinine and cystatin-C based equations as recently suggested by Andrew S. Levey [Ann Intern Med. 2020 Nov 10. doi: 10.7326/M20-6983].
We hope that the reviewer might find this additional background information helpful to complement his initial impressions and to follow our conclusions presented in the manuscript. We apologize for not having been clear enough in the description of our concept in a first instance and refer to the respective paragraph in the discussion section (lines 378f).
Second, the authors use CKD-EPI as reference equation, but a new equation was recently described, European Kidney Function Consortium (EKFC) equation, and look similar to their results in term of p30. In my opinion the authors must compared their results to this new equations, to really confirm the interest of their approach. the reference of the article is https://doi.org/10.7326/M20-4366
As the publication of the EKFC equation was only very recently and its critical appraisal by guideline committees is not yet finished, we originally had restricted our benchmarking to the recommended CKD-EPI equations and the updated Schwartz Bedside equation. Encouraged by the reviewer’s comment we now also provide a direct comparison with the EKFC equation. We revised the abstract, methods, results, and reference sections accordingly, added 97.5% confidence intervals for all performance relevant results and in this course corrected minor numerical deviations for CKD-EPI 2012 creatinine plus Cystatin-C equation. In addition, we divided original figure 2 in a revised figure 2 (creatinine only eGFR equations) and a revised figure 3 (eGFR equations using cystatin c). Original figure 3 is now revised figure 4. As our concept exceeded the performance of the EKFC equation, we feel that these results helped us to substantiate our concept further.
Reviewer 2 Report
The authors present the development and testing of a metabolite based serum test for accurate estimation of GFR in pediatric, adult, and geriatric patients with the potential of metabolic phenotyping of CKD patients.
It is novel interesting approach.
Is there is a patent pending - is the patent related to this findings?
Authors are employes of numares company - is the company involved in the development and later commercialization?
The answers to previous questions should be answered in the text to reduce potential commercial biases.
The study includes also children participants. Is the ethical issue and approval for them handled in all institutions? Some samples were generously gifted, are those approved by IRBs or ethical committees?
NMR is not available in all hospitals. How in practice the methodology will be made available widely? Commercialization of the test?
Author Response
The authors present the development and testing of a metabolite based serum test for accurate estimation of GFR in pediatric, adult, and geriatric patients with the potential of metabolic phenotyping of CKD patients.
It is novel interesting approach.
Is there is a patent pending - is the patent related to this findings?
As already stated in section 5 of our manuscript (revised lines 431f), the authors TS, JF, and ES have a patent application (DE-216820.2) pending.
Authors are employes of numares company - is the company involved in the development and later commercialization?
numares is a diagnostics company and focuses on the discovery, development and commercialization of diagnostic tests, using metabolite constellations to solve unmet medical needs that could not be addressed by single biomarker-based medicine. For further information, we also would like to refer the reviewer to the company website (https://www.numares.us/en).
As pointed out above, numares very successfully sells its NMR-based biomarkers in the US and is currently seeking to broaden and deepen its product pipeline further with developmental work in various medical fields, including nephrology, oncology and neurology.
The answers to previous questions should be answered in the text to reduce potential commercial biases.
We revised our conflict of interest section in lines 443f as follows:
Conflicts of Interest: TS, JF, SHo and ES report personal fees from numares AG, outside the submitted work. numares AG is a diagnostics company and focuses on the discovery, development and commercialization of diagnostic tests by metabolite constellations. JE serves as a scientific advisor for numares AG and receives financial compensation. All other authors have declared that no conflict of interest exists.
The study includes also children participants. Is the ethical issue and approval for them handled in all institutions? Some samples were generously gifted, are those approved by IRBs or ethical committees?
We revised the respective section lines 80f as follows:
For biomarker quantification bio-banked serum samples from 320 individuals from Lyon, Gothenburg, and Berlin [34] were used. All adult individuals gave informed consent before undergoing GFR measurement. As children were involved in this research activity, their assent and the permission of their parents was obtained. Assent was defined as a child's affirmative agreement to participate in research. A signed informed consent form from the child as well as from the parents was obtained. The respective institutional review boards covered ethical approval for NMR analysis of all the samples in adherence to the Declaration of Helsinki.
NMR is not available in all hospitals. How in practice the methodology will be made available widely? Commercialization of the test?
We would like to refer to our answer to the respective question of reviewer 1.
Round 2
Reviewer 2 Report
The authors sufficiently replied the questions and made the changes in the text.